**Subject Category:**
Biology (whole organism)

behaviour

cooperative behaviour, symbiosis, eusocial insect, *Aphaenogaster*, Madagascar

**Author for correspondence:**
Teppei Jono
e-mail: mjusinondo@gmail.com

†Present address: Tropical Biosphere Research Center, University of the Ryukyus, Senbaru 1, Nishihara, Nakagami-gun, Okinawa 903-0213, Japan.

# Novel cooperative antipredator tactics of an ant specialized against a snake

Teppei Jono[1,†], Yosuke Kojima[2] and Takafumi Mizuno[3]

[1]Department of Zoology, Graduate School of Science, Kyoto University, Sakyo, Kyoto 606-8502, Japan
[2]Department of Biology, Faculty of Science, Toho University, Miyama 2-1-1, Funabashi, Chiba 24-8510, Japan
[3]CAS Key Laboratory of Tropical Forest Ecology, Xishuangbanna Tropical Botanical Garden, Chinese Academy of Sciences, Menglun, Mengla 666303, Yunnan, People's Republic of China

TJ, 0000-0002-7274-7790

Eusocial insects can express surprisingly complex cooperative defence of the colony. Brood and reproductive castes typically remain in the nest and are protected by workers' various antipredator tactics against intruders. In Madagascar, a myrmicine ant, *Aphaenogaster swammerdami*, occurs sympatrically with a large blindsnake, *Madatyphlops decorsei*. As blindsnakes generally specialize on feeding on termites and ants brood by intruding into the nest, these snakes are presumably a serious predator on the ant. Conversely, a lamprophiid snake, *Madagascarophis colubrinus*, is considered to occur often in active *A. swammerdami* nests without being attacked. By presenting *M. colubrinus*, *M. decorsei* and a control snake, *Thamnosophis lateralis*, at the entrance of the nest, we observed two highly specialized interactions between ants and snakes: the acceptance of *M. colubrinus* into the nest and the cooperative evacuation of the brood from the nest for protection against the ant-eating *M. decorsei*. Given that *M. colubrinus* is one of the few known predators of blindsnakes in this area, *A. swammerdami* may protect their colonies against this blindsnake by two antipredator tactics, symbiosis with *M. colubrinus* and evacuation in response to intrusion by blindsnakes. These findings demonstrate that specialized predators can drive evolution of complex cooperative defence in eusocial species.

## 1. Background

Eusocial insects can express surprisingly variable and complex cooperative behaviours [1]. Their adult colony members are divided into reproductive and non-reproductive (or at least less-reproductive) castes. The latter caste cooperatively cares for the brood, forages, builds the nest and defends the colony, contributing to their success by enhancing colony fitness [2].

The understanding of the evolution of eusociality requires knowledge of the ecological causes of complex cooperative behaviours exhibited by the non-reproductive castes.

Predation risk is one of the major ecological factors selecting for group living, including the extreme of eusociality and effective antipredator tactics. Many vertebrates and invertebrates consume numerous eusocial insects [3–5]; thus, the vulnerable members of colonies such as brood and reproductive castes often remain in nests, which are isolated from enemy-rich space. Brood and reproductive castes are cooperatively protected by a number of colony workers that attack intruders by various antipredator behaviours such as clicking, biting, stinging, venom spraying and self-destructive defence [6,7]. The colonies of a myrmicine ant distributed in Madagascar, *Aphaenogaster swammerdami*, include approximately 100–1500 workers and inhabit large, underground nests that have one large entrance hole and a conspicuous mound [8,9]. This ant occurs sympatrically with a large blindsnake, *Madatyphlops decorsei*. Blindsnakes generally specialize in feeding on termites and ants brood inside their nests [10–12]. A single individual of blindsnakes can sometimes contain up to a thousand of prey items in its stomach [13]. Thus, this blindsnake, which is the largest blindsnake in Madagascar, is also presumably a serious predator of ant broods, excluding adults (workers). In addition to the ant predator *M. decorsei*, a lamprophiid snake, *Madagascarophis colubrinus*, which preys on various vertebrates [14,15], including a case of unknown species of blindsnakes (A. Mori, 2012, unpublished data), is often found in the active nests of ants. Therefore, it is called 'Ant's Mother' by local people of Madagascar [14–17]. Because both the 'Ant's Mother' and the blindsnake are likely to frequently enter the nest, and the latter can inflict intense predation pressure on the ant brood, we hypothesized that ant workers discriminate between and respond differently to predatory and non-predatory snake species. To test the discriminatory ability of ants and their responses to snake species with different ecological relationships, predator or symbiont, we first examined stomach contents of preserved specimens of *M. decorsei* to confirm predation of *A. swammerdami* by the snake. Subsequently, we presented *M. decorsei*, *M. colubrinus* and *Thamnosophis lateralis* (a sympatric frog-eating snake as control) at the entrance of *A. swammerdami* nest and compared the responses of ants to them.

# 2. Material and methods

## 2.1. Examination of stomach contents

Stomach contents of *M. decorsei* were examined in two preserved specimens in the collection of the Kyoto University Museum (Kyoto, Japan; KUZR 070527 and 079434). We identified the genus, caste and stage (queens, males, eggs, larvae and pupae) of each prey item.

## 2.2. Snake presentation experiment

The experiment was conducted in February 2013, March 2014 and February 2015 at Ankarafantsika National Park, Madagascar (16°18′ S, 46°48′ E). Nineteen nests of ants were found in the forest floor and more than five worker ants were collected from each nest for species identification. Seven *M. decorsei* (sex unknown, mean snout–vent length (SVL) = 300.0 (223–433) mm), seven *M. colubrinus* (four females and three males, mean SVL = 740.0 (610–994) mm) and five *T. lateralis* (two females and three males, mean SVL = 500.8 (467–510) mm) were collected and used for the experiment. Blindsnakes were collected in a pitfall trap and kept in a plastic cage with sand collected from its natural habitat. The other snakes were collected by hand in the forest both in the day and at night and kept in mesh bags. All snakes were released at the site of capture after the experiment.

A digital video camera (HDR-CX560 or DCR-SR220, Sony, Japan) was placed 1 m above the nest to record a 35 cm diameter field of view with the nest entrance at the centre. We only observed the behaviour of ants within the field of view as it would be a challenge to analyse the behaviour of all ants. To evaluate baseline activity, ant activity was recorded for 4 min prior to the presentation of snakes. Afterwards, a snake was presented into the nest while holding the posterior half of the body by hand, so that the head of the snake was at approximately 10 cm under the ground from the nest entrance. Five minutes after the presentation, the recording was terminated. During the experiment, the observers remained motionless and maintained more than 1 m distance from the nest by extending the hand holding the body of the snake to minimize disturbance to ants. The three snake species were presented to each nest in randomized order. In cases when the nest was abandoned or colony activity was obviously diminished before all the snake species had been presented, we terminated the experiment using that nest. In some cases, we could

not maintain all three species of snakes for an adequate duration. Therefore, of the three snake species, we presented only two to an ant nest in 5 out of 19 cases and only 1 to a nest in 6 cases. In total, 11, 15 and 14 trials were conducted using *M. decorsei*, *M. colubrinus* and *T. lateralis*, respectively. Intervals between consecutive trials for a single ant nest were more than 1 day to minimize carry-over effects.

## 2.3. Data analysis

Based on the analyses of the digital video recordings, ant behaviour was classified as follows: normal activity, alert, biting and nest evacuation with a larva or pupa. Alert behaviour was defined as running at a perceptively more rapid pace (greater than $5 \text{ cm s}^{-1}$) compared with normal activity (approx. $1 \text{ cm s}^{-1}$). Biting was defined as biting snake body using the mandibles, while evacuation behaviour was defined as emergence from the nest entrance carrying a larva or pupa in the mandibles. Behaviour other than the aforementioned three were included under normal activity. To compare reactions of ants among the snake stimuli, the total number of ants engaged in biting when a snake was presented was compared by the generalized linear mixed models using a Poisson error distribution and a log link (GLMM; R package lme4), treating nest identity and snake identity as random effects [18]. The statistical significance of effects was investigated by *post hoc* tests with the Tukey style contrasts for pairwise comparisons of snake species (R package multcomp) [19]. To evaluate the immediate reaction of ants towards the snakes, the number of all the ants on screen was counted every 5 s, from 20 s before the presentation of the snake to 60 s after the presentation. The number of ants was compared between the 15 s before and after the snake presentation, using the Wilcoxon signed-rank test. To evaluate the sustained reaction of ants towards the snakes, the number of ants engaged in alert or evacuation behaviour was counted during every 1 min from 4 min before the presentation of the snake until 5 min after the presentation. The number of ants engaged in each behaviour was counted every 1 min. The numbers of ants engaged in these behaviours 4 min before and after the presentation were compared by the Wilcoxon signed-rank test. The full dataset used for analyses in this study has been uploaded as part of the electronic supplementary material. All statistical tests were performed using R 3.1.1 [20].

## 3. Results

All stomach contents of *M. decorsei* were identified as *Aphaenogaster* pupae (51/51 from KUZR070527 and 10/10 from KUZR079434 of identifiable prey items). The worker ants exhibited an aggressive reaction towards *T. lateralis*, no reaction towards *M. colubrinus* (figure 1*a*) and immediate cooperative evacuation of pupae and larvae in response to *M. decorsei* (figure 1*b*; electronic supplementary material, movie S1). Significant differences in the number of biting ants were detected in *M. decorsei* versus *M. colubrinus* ($p = 0.014$) and *T. lateralis* versus *M. colubrinus* ($p < 0.001$), but not in *M. decorsei* versus *T. lateralis* ($p = 0.316$; table 1 and figure 1*c*). The analysis was not blindly performed; biting and evacuation were clearly distinguishable from other behaviours based on the digital video recordings. However, the effect size may have been influenced by a potential bias. Attacking behaviour other than biting was not observed during the experiment. Workers exhibited cooperative evacuation behaviour in all trials using *M. decorsei*, and only after the presentation of the blindsnake; therefore, we did not conduct a statistical test to compare evacuation behaviour among snake species (figure 1*c*).

Upon the presentation of *M. decorsei*, most worker ants immediately ran into the nest, resulting in a rapid decrease in the number of worker ants ($p = 0.012$ in the 15 s before versus after presentation); this phenomenon was not observed upon the presentation of *T. lateralis* and *M. colubrinus* ($p = 0.447$ and $0.519$ in the 15 s before versus after presentation; figure 2*a*). The worker ants showed alert behaviour (clearly running at a faster speed, greater than $5 \text{ cm s}^{-1}$) against *M. decorsei* and *T. lateralis*, but rarely in response to *M. colubrinus* ($p = 0.946$ in *M. decorsei* versus *T. lateralis*, $p < 0.001$ in both *M. decorsei* versus *M. colubrinus* and *T. lateralis* versus *M. colubrinus*; figure 2*b*). In the case of *M. decorsei*, the worker ants showed both alert and evacuation behaviours following the presentation of the snake ($p < 0.001$ comparing alert behaviour 4 min before versus after presentation; no statistical test was conducted on evacuation behaviour because no ant exhibited the behaviour before the presentation; figure 2*b*). The number of workers that showed alert behaviour was largest 0–1 min after the presentation, followed by evacuation behaviour with a larva or pupa 2–5 min after the presentation. A number of workers, which was more than those that had entered the nest, emerged from the entrance carrying larvae or pupae and ran to the top of adjacent vegetation. No workers carried eggs, males or queens. Part of them ran out of filming range; therefore, the workers were not included in subsequent analysis. The evacuation did not result in permanent relocation of the

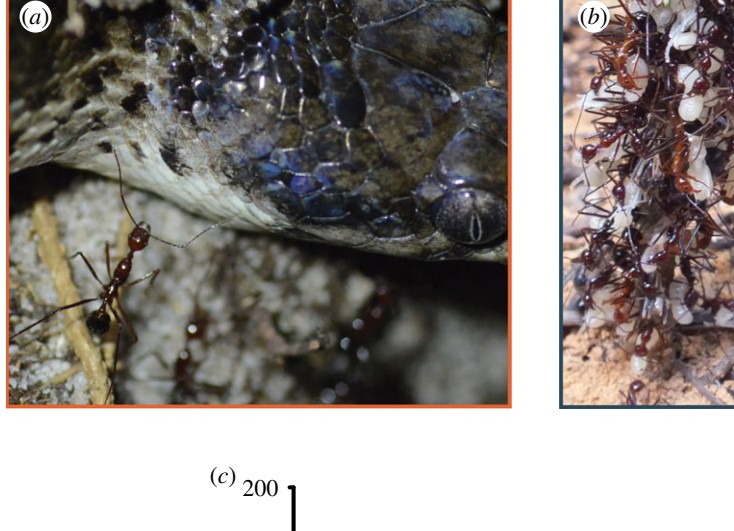

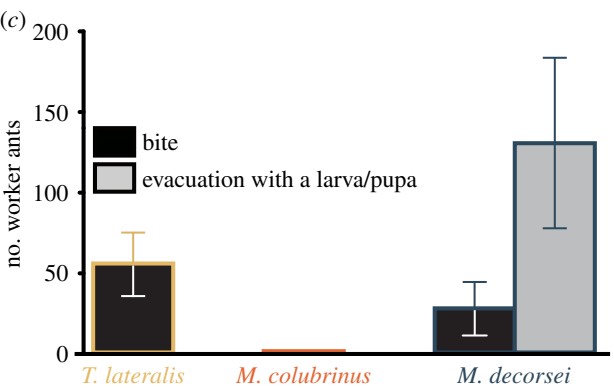

**Figure 1.** Reaction of *A. swammerdami* towards *M. colubrinus* (*a*) and *M. decorsei* (*b*). The workers that reacted to *M. decorsei* were standing on the vegetation adjacent to the nest entrance. (*c*) Comparison of the number of worker ants (mean ± s.e.) exhibiting biting or evacuation with brood.

**Table 1.** Result of the pairwise comparisons of the effects of the snake species on number of worker ants exhibiting biting based on generalized linear mixed model using the Tukey *post hoc* test. TL, MC and MD indicate frog-eating *T. lateralis*, vertebrate-eating *M. colubrinus* and ant-eating *M. decorsei*, respectively.

| random effects | variance | | | |
|---|---|---|---|---|
| nest identity | 1.438 | | | |
| snake identity | 2.612 | | | |
| fixed effect | estimates | s.e. | z | *p*-value |
| snake species | | | | |
| TL-MC | 4.418 | 1.082 | 4.082 | <0.001 |
| TL-MD | 1.496 | 1.033 | 1.449 | 0.316 |
| MC-MD | 2.922 | 1.043 | 2.800 | 0.014 |

nest, but was a temporary move; 10 of 11 colonies recovered normal activity 1 day after the *M. decorsei* trials, without abandoning the nest. When presented with *T. lateralis*, the worker ants showed alert behaviour ($p < 0.001$ in the 4 min before versus after presentation), whereas few ants showed alert behaviour when presented with *M. colubrinus* ($p = 0.162$ in the 4 min before versus after presentation; figure 2*b*).

## 4. Discussion

The present results demonstrate discriminatory ability of *A. swammerdami* ants towards snake species. In addition, we found two highly specialized interactions between the ants and the snakes: (i) the acceptance of *M. colubrinus* entering the nest and (ii) a cooperative evacuation of the brood from

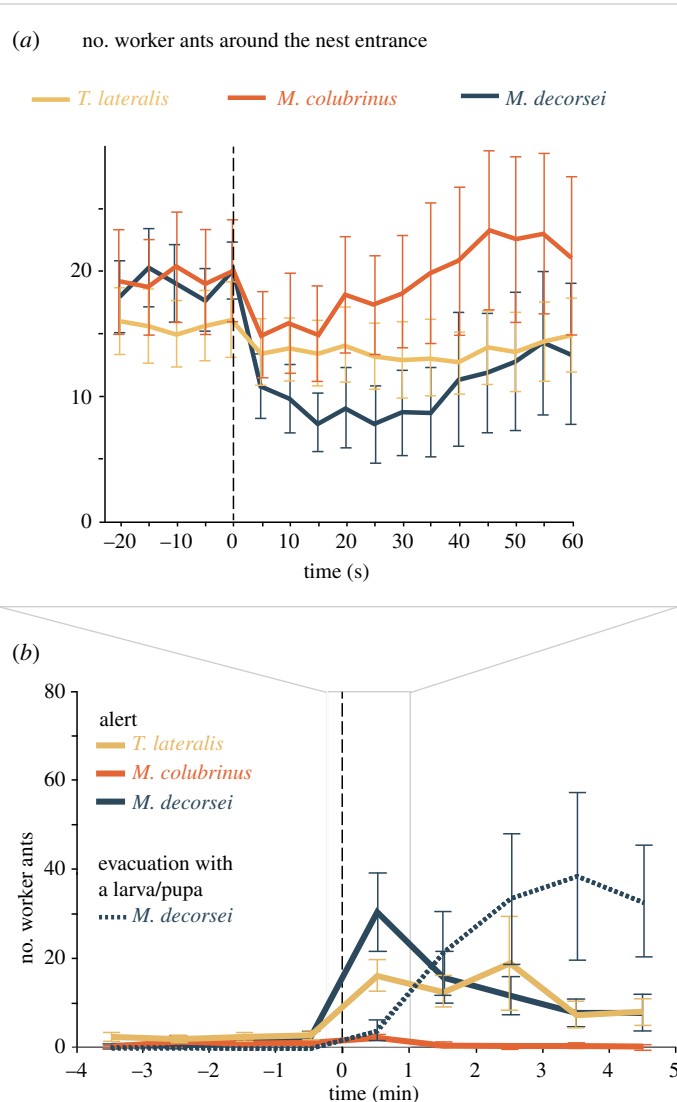

**Figure 2.** (a) Changes in the number of worker ants (mean ± s.e.) around nest entrance from 20 s before to 60 s after the presentation of a snake. (b) Changes in the number of worker ants (mean ± s.e.) that showed alert behaviour or evacuation with a larva or pupa from 4 min before to 5 min after the presentation of a snake.

the nest to protect against ant-eating *M. decorsei*. The limited data on stomach contents of *M. decorsei* demonstrate that it feeds on pupae of *Aphaenogaster*. Although no ant larvae were found in the two stomachs examined, this blindsnake presumably also preys on larvae as other blindsnakes [10–13]. Note that the more rapid digestion of soft-bodied larvae compared with pupal exoskeletons, which were retained, may account for the non-existence of larvae in the examined stomachs. Overall, it is likely that the cooperative transport of larvae and pupae of *A. swammerdami* from the nest represents evacuation of ants in these life-history stages in the face of predation risk from the blindsnake, moving them to the ground surface where they face a higher risk of attack by predators other than blindsnakes. Because nest evacuation was performed solely against *M. decorsei*, this behaviour in *A. swammerdami* would be a highly specialized antipredator tactic against *M. decorsei* and perhaps other typhlopid snakes.

Although several vertebrates, including anteaters, black woodpeckers, horned lizards and blindsnakes, are known to feed specifically on ants, no specialized antipredator tactics by ants towards vertebrate predators have been previously reported. To the best of our knowledge, this is the first report of such predator-specific defensive behaviour by ants in response to vertebrates. Several studies have reported similar behaviour by some genera of American ants, including *Aphaenogaster*, when they are attacked by army ants [21–25]. The victimized ants cooperatively evacuate with their brood several minutes after contact with army ants, which is consistent with our findings. They return several hours after the exodus, suggesting their behaviour also is a temporary evacuation [22]. Because no army ants occur in Madagascar [26], we suggest that cooperative evacuation of brood by *A. swammerdami* represents a

specialized defence against blindsnakes. Considering that typhlopid snakes are also distributed in the New World, American *Aphaenogaster* might exhibit cooperative evacuation not only against army ants but also against blindsnakes. Although most of ant predators mainly forage at the nest entrance to probably facilitate cooperative biting as an effective antipredator tactic against intrusion, both blindsnakes and army ants are capable of entering the nest and often prey on a large number of individuals within a single foraging bout [10,25]. In addition, blindsnakes may have specialized in feeding on ant broods in the nest. Shifting from a defence strategy to the evacuation of ant broods outside the nest against such serious and specialized predators, therefore, is potentially beneficial for colony survival. The specialized predation risk would drive the evolution of the specialized antipredator behaviour, in turn, a complex cooperative defence system in ants. It would be interesting to examine whether nest evacuation against blindsnakes has independently evolved from evacuation against army ants or whether these two behavioural responses share the same evolutionary origin.

*Aphaenogaster swammerdami* attacked *T. lateralis* by biting, suggesting that the ants typically attack nest intruders aggressively. Nevertheless, the ant showed virtually no aggressive response towards *M. colubrinus*. After the antennal palpation of the snake, *A. swammerdami* returned to normal activity without attacking it. A few previous studies have reported snake–ant symbioses in North America; however, the reported cases only occur during the non-reproductive season of the ants when they are inactive [27]. The utilization of ant nests is likely to be beneficial for snakes as retreats or nesting sites, as ant nests generally provide relatively constant temperature and humidity conditions [28]. This would be particularly true for snakes at our study site, where they experience six months of severe dry conditions [29,30]. Symbiosis with *M. colubrinus* may also be beneficial for the ant because this snake is one of the few known predators of blindsnakes in this area, suggesting that *M. colubrinus* can protect the ants' brood from the predatory *M. decorsei*. In fact, symbiosis between *M. colubrinus* and *A. swammerdami* may not be the only case of snake–ant symbiosis in Madagascar. Another reptile-eating snake, *Leioheterodon modestus* was sometimes observed from the nest entrance of *A. swammerdami* [14,17], although no experimental evidence exists to support their symbiosis. Notably, another species of a Madagascan ant, *Camponotus imitator*, also shares its nests with *M. colubrinus* and *L. modestus* [31]. The association with reptile-eating snakes could be another antipredator tactic of ants against blindsnakes. However, it is also plausible that these snakes chemically camouflage to avoid attack by the ant similar to that observed in the case of the African frog coexisting with aggressive ants [32]. Future investigation is required to test whether the unusual relationship between ants and snakes in Madagascar represents mutualism or commensalism.

Ethics. We conducted all experiments in compliance with the guidelines of the Animal Care and Use Committee of Kyoto University. Regarding the reduction in sample size, we tried to use the minimum number of animals necessary to achieve the research objectives in the experiment. Permission to conduct the research at the field site was granted by Ministere de l'Environnement et des Forêts (268/13/MEF/SG/DGF/DCB.SAP/SCB for 2013, 235/14/MEEF/SG/DGF/DCB.SAP/SCB for 2014 and 124/15/MEEMF/SG/DGF/DCB.SAP/SCBT for 2015).

Data accessibility. The full dataset used for analyses in this study has been uploaded as part of the electronic supplementary material.

Authors' contributions. T.J., Y.K. and T.M. designed research; T.J. and Y.K. collected behavioural data; T.M. examined stomach contents; T.J. analysed data and wrote paper. All authors gave final approval for publication.

Competing interests. We declare we have no competing interests.

Funding. This work was financially supported in part by Grants-in-Aid for International Scientific Research (B) (nos. 17405007 and 24405008) from the Ministry of Education, Culture, Sports, Science, and Technology, Japan (MEXT); Grant-in-Aid for JSPS Research Fellow (no. 231388) and Grant-in-Aid for Young Scientists (B) (no. 17K17970) from MEXT to T.J.; Shikata Memorial Trust for Nature Conservation to Y.K.; and a Grant for Basic Science Research Projects from the Sumitomo Foundation (no. 131168) to Y.K.

Acknowledgements. We are grateful to A. Mori for support in fieldwork, numerous comments on the research and reviewing the manuscript. We also thank Madagascar National Parks for providing facilities in Ankarafantsika National Park and for allowing us to collect animals and conduct the behavioural experiments; H. Numata for valuable comments on the research; A.H. Savitzky for helpful comments on this study and for reviewing the manuscript; F. Rakotondraparany, H. Rakotomanana for support in fieldwork; M. Hori, H. Sato and B. Razafimahatratra for providing information about the ants and snakes; and H.R. Maheritafika and J.M.L. Rakotoarison for collection of the snakes.

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
