## [Reviewer comments · Royal Society Open Science]

Review History

RSOS-190283.R0 (Original submission)

Review form: Reviewer 1

Is the manuscript scientifically sound in its present form?

Yes

Are the interpretations and conclusions justified by the results?

Yes

Is the language acceptable?

Yes

Is it clear how to access all supporting data?

Yes

Do you have any ethical concerns with this paper?

No

Have you any concerns about statistical analyses in this paper?

No

Recommendation?

Accept with minor revision (please list in comments)

Comments to the Author(s)

This is an excellent experimental study conclusively demonstrating behavioral adaptation of a Malagasy ant species to a predator, and hinting at a complex possible symbiosis with another species of snake that may help to deter the predatory blindsnake.

The frequent occurrence of *Madagascarophis* (and a few other snakes) in ant nests in wetlands in Madagascar is a long-known mystery, and this is the first paper providing a biologically sound explanation with experimental evidence for it. I greatly enjoyed reading it, and strongly recommend its publication.

Note that in my reviews, I very often provide long lists of comments, suggestions and corrections, but in this case I could find only few points that warrant consideration during revision, all minor. Overall this is an excellent study!

Lines 55-56: Something is wrong with this sentence. Please rephrase.

Line 152: In how many cases (of N=11 in this treatment) was this behavior observed?

Lines 195-196: Higher risk of predation – but of other predators, not the blindsnake, right? Please rephrase to be more clear.

Lines 240-242: Please rephrase to make more clear that this idea of additional ant-snake symbioses in Madagascar is a hypothesis you are proposing here, but no experimental (and I think not even anecdotal?) evidence exists so far to support it.

Line 258: "de l'Environnement et des Forêts"

Line 266: We

Review form: Reviewer 2 (Elva J H Robinson)**Is the manuscript scientifically sound in its present form?**

Yes

Are the interpretations and conclusions justified by the results?

Yes

Is the language acceptable?

No

Is it clear how to access all supporting data?

Yes

Do you have any ethical concerns with this paper?

Yes

Have you any concerns about statistical analyses in this paper?

No

Recommendation?

Accept with minor revision (please list in comments)

Comments to the Author(s)

Overall, I think this is an interesting study which makes a valuable and novel contribution to the literature. The authors show that ants (*Aphaenogaster swammerdami*) respond differently to 3 snake species: an ant-eating snake (*Madatyphlops decorsei*), a frog-eating snake (*Thamnosophis lateralis*), and a snake-eating snake (*Madagascarophis colubrinus*). Specifically, the ants respond to the ant-eating snake with alertness and evacuation of brood, to the frog-eating snake with some agonistic behaviour, but do not show any significant agonism towards the snake-eating snake. This suggests that not only have the ants evolved direct anti-predator responses (responding selectively to the ant-eating snake), but they have also evolved to not repel snakes that could indirectly protect them from predators (by accepting the snake-eating snake). The experiments are well-thought out and the discussion is appropriate to the results presented, and raises some interesting questions about the evolution of ant-snake symbioses.

My concerns with this paper are about clarity and methodology, and can be addressed in a revision.

Abstract & Introduction:

In the abstract it is not clear what the roles of the three snakes are. It is clear that the blindsnake is a predator, and the *Thamnosophis* snake is a control, but it is not clear what the role of the lamprophiid snake is, it is simply referred to as occurring in active nests. The key point, its predation on the blindsnake, is not mentioned until the end of the abstract.

In the introduction, it becomes clear that the lamprophiid snake's role in the study is as a contrasting predator of the first snake - but in this section the control snake is not mentioned at all, making it hard to interpret the results and work out which comparisons are the important ones. This just needs to be clearer.

Methods.

Line 100. How was activity quantified? Or do you just mean the entrance to the nest was recorded, i.e. video recorded? Please make this clearer

Lines 117-124. Was extraction from the videos performed blind to the hypothesis? behaviours such as alertness and biting can be subjective to determine, risking unconscious observer bias if the hypothesis is known. As the analysis was done from video data, there are no logistic restrictions to prevent the data extraction being carried out blind to the identity of the snake, i.e. by a person unfamiliar with the hypothesis of the experiment and the various snake species. If the analysis has been performed blind, please add a note to this effect. If it has not been performed blind, the authors should address this by providing more information about the behavioural criteria used, and also discuss their effect sizes in the context of possible bias, or they should reanalyse a subset of their videos blind to check for observer bias effects.

Line 124. Was the baseline activity level taken into consideration in these models? Please add a table of the model output.

Minor points:

Line 129. Number of ants doing what? Is this all the ants on screen?

The paper would also benefit from thorough proof-reading for grammar. For example, in the first few lines of the abstract, line 17, "for" should read "of"; line 19 "worker's" should read "workers' ", line 22 "in" should read "on"; "termites" should read "termites' ", etc. I have not commented on other typographical and grammatical errors throughout the text.

Decision letter (RSOS-190283.R0)

13-Jun-2019

Dear Professor Jono

On behalf of the Editors, I am pleased to inform you that your Manuscript RSOS-190283 entitled "Novel Cooperative Antipredator Tactics of an Ant Specialized against a Snake" has been accepted for publication in Royal Society Open Science subject to minor revision in accordance with the referee suggestions. Please find the referees' comments at the end of this email.

The reviewers and handling editors have recommended publication, but also suggest some minor revisions to your manuscript. Therefore, I invite you to respond to the comments and revise your manuscript.

- Ethics statement

- Data accessibility

<http://datadryad.org/submit?journalID=RSOS&manu=RSOS-190283>

- Competing interests

- Authors' contributions

All submissions, other than those with a single author, must include an Authors' Contributions section which individually lists the specific contribution of each author. The list of Authors should meet all of the following criteria; 1) substantial contributions to conception and design, or

acquisition of data, or analysis and interpretation of data; 2) drafting the article or revising it critically for important intellectual content; and 3) final approval of the version to be published.

- Acknowledgements

- Funding statement

Because the schedule for publication is very tight, it is a condition of publication that you submit the revised version of your manuscript before 22-Jun-2019. Please note that the revision deadline will expire at 00.00am on this date. If you do not think you will be able to meet this date please let me know immediately.

- 1) A text file of the manuscript (tex, txt, rtf, docx or doc), references, tables (including captions) and figure captions. Do not upload a PDF as your "Main Document";

- 2) A separate electronic file of each figure (EPS or print-quality PDF preferred (either format should be produced directly from original creation package), or original software format);
- 3) Included a 100 word media summary of your paper when requested at submission. Please ensure you have entered correct contact details (email, institution and telephone) in your user account;
- 4) Included the raw data to support the claims made in your paper. You can either include your data as electronic supplementary material or upload to a repository and include the relevant doi within your manuscript. Make sure it is clear in your data accessibility statement how the data can be accessed;
- 5) All supplementary materials accompanying an accepted article will be treated as in their final form. Note that the Royal Society will neither edit nor typeset supplementary material and it will be hosted as provided. Please ensure that the supplementary material includes the paper details where possible (authors, article title, journal name).

If your manuscript is newly submitted and subsequently accepted for publication, you will be asked to pay the article processing charge, unless you request a waiver and this is approved by Royal Society Publishing. You can find out more about the charges at <http://rsos.royalsocietypublishing.org/page/charges>. Should you have any queries, please contact openseience@royalsociety.org.

Kind regards,
Alice Power
Editorial Coordinator
Royal Society Open Science
openseience@royalsociety.org

on behalf of Dr Ryan Earley (Associate Editor) and Kevin Padian (Subject Editor)
openseience@royalsociety.org

Reviewer comments to Author:
Reviewer: 1

Comments to the Author(s)

This is an excellent experimental study conclusively demonstrating behavioral adaptation of a Malagasy ant species to a predator, and hinting at a complex possible symbiosis with another species of snake that may help to deter the predatory blindsnake.

The frequent occurrence of *Madagascarophis* (and a few other snakes) in at nests in wetsren Madagascar is a long-known mystery, and this is the first paper providing a biologically sound explanation with experimental evidence for it. I greatly enjoyed reading it, and strongly recommend its publication.

Note that in my reviews, I very often provide long lists of comments, suggestions and corrections, but in this case I could find only few points that warrant consideration during revision, all minor. Overall this is an excellent study!

Lines 55-56: Something is wrong with this sentence. Please rephrase.

Line 152: In how many cases (of N=11 in this treatment) was this behavior observed?

Lines 195-196: Higher risk of predation – but of other predators, not the blindsnake, right? Please rephrase to be more clear.

Lines 240-242: Please rephrase to make more clear that this idea of additional ant-snake symbioses in Madagascar is a hypothesis you are proposing here, but no experimental (and I think not even anecdotal?) evidence exists so far to support it.

Line 258: "de l'Environnement et des Forêts"

Line 266: We

Reviewer: 2

Comments to the Author(s)

Overall, I think this is an interesting study which makes a valuable and novel contribution to the literature. The authors show that ants (*Aphaenogaster swammerdami*) respond differently to 3 snake species: an ant-eating snake (*Madatyphlops decorsei*), a frog-eating snake (*Thamnosophis lateralis*), and a snake-eating snake (*Madagascarophis colubrinus*). Specifically, the ants respond to the ant-eating snake with alertness and evacuation of brood, to the frog-eating snake with some agonistic behaviour, but do not show any significant agonism towards the snake-eating snake. This suggests that not only have the ants evolved direct anti-predator responses (responding selectively to the ant-eating snake), but they have also evolved to not repel snakes that could indirectly protect them from predators (by accepting the snake-eating snake). The experiments are well-thought out and the discussion is appropriate to the results presented, and raises some interesting questions about the evolution of ant-snake symbioses.

My concerns with this paper are about clarity and methodology, and can be addressed in a revision.

Abstract & Introduction:

In the abstract it is not clear what the roles of the three snakes are. It is clear that the blindsnake is a predator, and the *Thamnosophis* snake is a control, but it is not clear what the role of the lamprophiid snake is, it is simply referred to as occurring in active nests. The key point, its predation on the blindsnake, is not mentioned until the end of the abstract.

In the introduction, it becomes clear that the lamprophiid snake's role in the study is as a contrasting predator of the first snake - but in this section the control snake is not mentioned at

all, making it hard to interpret the results and work out which comparisons are the important ones. This just needs to be clearer.

Methods.

Line 100. How was activity quantified? Or do you just mean the entrance to the nest was recorded, i.e. video recorded? Please make this clearer

Lines 117-124. Was extraction from the videos performed blind to the hypothesis? behaviours such as alertness and biting can be subjective to determine, risking unconscious observer bias if the hypothesis is known. As the analysis was done from video data, there are no logistic restrictions to prevent the data extraction being carried out blind to the identity of the snake, i.e. by a person unfamiliar with the hypothesis of the experiment and the various snake species. If the analysis has been performed blind, please add a note to this effect. If it has not been performed blind, the authors should address this by providing more information about the behavioural criteria used, and also discuss their effect sizes in the context of possible bias, or they should reanalyse a subset of their videos blind to check for observer bias effects.

l'm 124. Was the baseline activity level taken into consideration in these models? Please add a table of the model output.

Minor points:

Line 129. Number of ants doing what? Is this all the ants on screen?

The paper would also benefit from thorough proof-reading for grammar. For example, in the first few lines of the abstract, line 17, "for" should read "of"; line 19 "worker's" should read "workers' ", line 22 "in" should read "on"; "termites" should read "termites' ", etc. I have not commented on other typographical and grammatical errors throughout the text.

Author's Response to Decision Letter for (RSOS-190283.R0)

See Appendix A.

Decision letter (RSOS-190283.R1)

23-Jul-2019

Dear Professor Jono,

I am pleased to inform you that your manuscript entitled "Novel Cooperative Antipredator Tactics of an Ant Specialized against a Snake" is now accepted for publication in Royal Society Open Science.

Royal Society Open Science operates under a continuous publication model

(<http://bit.ly/cpFAQ>). Your article will be published straight into the next open issue and this will be the final version of the paper. As such, it can be cited immediately by other researchers. As the issue version of your paper will be the only version to be published I would advise you to check your proofs thoroughly as changes cannot be made once the paper is published.

on behalf of Dr Ryan Earley (Associate Editor) and Kevin Padian (Subject Editor)
openscience@royalsociety.org

Appendix A

Dear Reviewers and Editors,

Thank you for your thoughtful review of our manuscript. We have addressed your critiques to the best of our abilities. Your comments appear below in black font and we address them point by point in red font. Additionally, we highlighted the sentences that have been changed in the revised manuscript. We hope the revised manuscript will meet your approval for publication in Royal Society Open Science.

Sincerely, Teppei

Comments from Reviewer #1:

General:

This is an excellent experimental study conclusively demonstrating behavioral adaptation of a Malagasy ant species to a predator, and hinting at a complex possible symbiosis with another species of snake that may help to deter the predatory blindsnake.

The frequent occurrence of *Madagascarophis* (and a few other snakes) in ant nests in wetland Madagascar is a long-known mystery, and this is the first paper providing a biologically sound explanation with experimental evidence for it. I greatly enjoyed reading it, and strongly recommend its publication.

Note that in my reviews, I very often provide long lists of comments, suggestions and corrections, but in this case I could find only few points that warrant consideration during revision, all minor. Overall this is an excellent study!

Many thanks for the encouraging words.

> Lines 55-56: Something is wrong with this sentence. Please rephrase.

This has been changed to “The colonies of a myrmicine ant found in Madagascar, *Aphaenogaster swammerdami*, include approximately 100–1,500 workers inhabit large, underground nests that have one large entrance hole and a conspicuous mound”. See revised p. 4 lines 56-59.

> Line 152: In how many cases (of N=11 in this treatment) was this behavior observed?

This has been changed to “Workers exhibited cooperative evacuation behaviour in all

trials using *M. decorsei*, and only after the presentation of the blindsnake”. See revised p. 11 lines 170-171.

> Lines 195-196: Higher risk of predation – but of other predators, not the blindsnake, right? Please rephrase to be more clear.

This has been changed to “they face a higher risk of attack by predators other than blindsnakes”. See revised p. 14 lines 215-216.

> Lines 240-242: Please rephrase to make more clear that this idea of additional ant-snake symbioses in Madagascar is a hypothesis you are proposing here, but no experimental (and I think not even anecdotal?) evidence exists so far to support it.

We agree with your comment. We have added the sentence “although no experimental evidence exists to support their symbiosis”. See revised p. 18 lines 265-266.

> Line 258: "de l'Environnement et des Forêts"

We have changed “de L'environnement et des Forets” to “de l'Environnement et des Forêts”. See revised p. 19 line 280.

> Line 266: We

We have changed “I” to “We”. See revised p. 19 line 288.

Comments from Reviewer #2:

General:

Overall, I think this is an interesting study which makes a valuable and novel contribution to the literature. The authors show that ants (*Aphaenogaster swammerdami*) respond differently to 3 snake species: an ant-eating snake (*Madatyphlops decorsei*), a frog-eating snake (*Thamnosophis lateralis*), and a snake-eating snake (*Madagascarophis colubrinus*). Specifically, the ants respond to the ant-eating snake with alertness and evacuation of brood, to the frog-eating snake with some agonistic behaviour, but do not show any significant agonism towards the snake-eating snake. This suggests that not only have the ants evolved direct anti-predator responses (responding selectively to the ant-eating snake), but they have also evolved to not repel snakes that could

indirectly protect them from predators (by accepting the snake-eating snake). The experiments are well-thought out and the discussion is appropriate to the results presented, and raises some interesting questions about the evolution of ant-snake symbioses.

My concerns with this paper are about clarity and methodology, and can be addressed in a revision.

Sincerest thanks for your comments on our manuscript.

> In the abstract it is not clear what the roles of the three snakes are. It is clear that the blindsnake is a predator, and the *Thamnosophis* snake is a control, but it is not clear what the role of the lamprophiid snake is, it is simply referred to as occurring in active nests. The key point, its predation on the blindsnake, is not mentioned until the end of the abstract.

Thank you for your good advice. Although unknown species of blind snakes have been obtained from a stomach of *M. colubrinus* at our study site (A. Mori 2012, unpublished data), predation of our subject blind snake species, *M. decorsei*, by the snake-eating snake is yet to be confirmed. Therefore, the purpose of the present study was to investigate the discriminatory capacity of ants and their responses to snake species exhibiting different ecological interactions including predation or symbiosis. We proposed a potential mutualism between the snake-eating snake and the ants. Our future study would investigate the predation of *M. decorsei* by *M. colubrinus* and test whether the relationship between ants and snakes in Madagascar represents mutualism or commensalism. To clarify it, we have corrected abstract and introduction.

> In the introduction, it becomes clear that the lamprophiid snake's role in the study is as a contrasting predator of the first snake - but in this section the control snake is not mentioned at all, making it hard to interpret the results and work out which comparisons are the important ones. This just needs to be clearer.

We agree with you and have added the sentence “and *Thamnosophis lateralis* (a sympatric frog-eating snake as control)” and removed relevant sentences from materials & methods. See revised p. 5 lines 77-79 and p. 6 line 95-p. 7 line 99.

> Line 100. How was activity quantified? Or do you just mean the entrance to the nest was

recorded, i.e. video recorded? Please make this clearer

Because it is difficult to observe behaviour of all the ants, we only recorded and observed the behaviour of ants within the field of view of this video camera. To clarify we have added the sentence “We only observed the behaviour of ants within the field of view as it would be a challenge to analyse the behavior of all ants”. See revised p. 7 lines 106-108.

> Lines 117-124. Was extraction from the videos performed blind to the hypothesis? behaviours such as alertness and biting can be subjective to determine, risking unconscious observer bias if the hypothesis is known. As the analysis was done from video data, there are no logistic restrictions to prevent the data extraction being carried out blind to the identity of the snake, i.e. by a person unfamiliar with the hypothesis of the experiment and the various snake species. If the analysis has been performed blind, please add a note to this effect. If it has not been performed blind, the authors should address this by providing more information about the behavioural criteria used, and also discuss their effect sizes in the context of possible bias, or they should reanalyse a subset of their videos blind to check for observer bias effects.

We agree with you and have added more precise definition of ants' behaviour and comments on the effect sizes in the context of possible bias. See revised p. 8 line 129-p. 9 line 136 and p. 11 lines 166-169.

> l'm 124. Was the baseline activity level taken into consideration in these models? Please add a table of the model output.

We did not include the baseline activity to prevent the model becoming too complex. We have added the table to show output of the model. See added table 1.

> Line 129. Number of ants doing what? Is this all the ants on screen?

We have changed the sentence “the number of all the ants on screen”. See revised p. 9 lines 143-144.

> The paper would also benefit from thorough proof-reading for grammar. For example, in the first few lines of the abstract, line 17, "for" should read "of"; line 19 "worker's" should read "workers' ", line 22 "in" should read "on"; "termites" should read "termites' ", etc. I have not commented on other typographical and grammatical errors throughout the text.

Thank you for your comments. We have made the corrections as suggested. The English in this document has been checked by at least two professional editors, both native

speakers of English.

In addition, we have addressed following matters pointed out when the manuscript was unsubmitted.

> -- Data Availability --

As your paper contains multiple ESM files, please update your data availability statement to specify which data are uploaded as supplementary material. For example: "The XXXX video file and XXXX data are uploaded as supplementary material".

We have made the corrections as suggested. See revised p. 19 line 285.

> -- Ethics --

Please expand your Ethics statement to comply with journal policies. We note that: "we could not maintain all three

120 species of snakes for an adequate duration,"; please expand upon this statement, and clarify the fate of these snakes. Authors should include details of animal welfare (such as species, number, gender, age, weight, housing conditions, welfare, training and the fate of ALL animals at the end of the experiment) and relevant details of steps taken to ameliorate suffering.

We agree with you and have added more detail explanation. See revised p. 8 lines 120-122.

> -- ESM Titles and Legends --

Your manuscript was unsubmitted because unfortunately the headings and/or legends you have provided for your supplementary files are not sufficiently informative. Supplementary files will be published alongside the paper on the journal website and posted on the online figshare repository (for Royal Society Publishing examples, please see <https://rs.figshare.com/>). By making ESM available on the figshare website, we hope to increase the visibility of your data, thus ensuring it is available for and to the benefit of a wide audience.

The heading and legend provided for each supplementary file during the submission process will be used to create the figshare page, so please ensure these are accurate and informative so that your files can be found in searches (ie: the titles should not be given only as "Supplementary X" within the ScholarOne submission form). Files on figshare will be made available approximately one week before the accompanying article so that the supplementary material can be attributed a unique DOI, thus allowing the data to be cited more easily.

We have added legends for supplementary files as suggested. See revised p. 30 lines 419-422.

Appendix B

Dear Reviewers and Editors,

Thank you for your thoughtful review of our manuscript. We have addressed your critiques to the best of our abilities. Your comments appear below in black font and we address them point by point in red font. Additionally, we highlighted the sentences have been changed in the revised manuscript. We hope the revised manuscript will meet your approval for publication in Royal Society Open Science.

Sincerely, Teppeï

Comments from Reviewer #1:

General:

This is an excellent experimental study conclusively demonstrating behavioral adaptation of a Malagasy ant species to a predator, and hinting at a complex possible symbiosis with another species of snake that may help to deter the predatory blindsnake.

The frequent occurrence of *Madagascarophis* (and a few other snakes) in ant nests in wetland Madagascar is a long-known mystery, and this is the first paper providing a biologically sound explanation with experimental evidence for it. I greatly enjoyed reading it, and strongly recommend its publication.

Note that in my reviews, I very often provide long lists of comments, suggestions and corrections, but in this case I could find only few points that warrant consideration during revision, all minor. Overall this is an excellent study!

Many thanks for the encouraging words.

> Lines 55-56: Something is wrong with this sentence. Please rephrase.

This has been changed to “The colonies of a myrmicine ant found in Madagascar, *Aphaenogaster swammerdami*, include approximately 100–1,500 workers inhabit large, underground nests that have one large entrance hole and a conspicuous mound”. See revised p. 4 lines 56-59.

> Line 152: In how many cases (of N=11 in this treatment) was this behavior observed?

This has been changed to “Workers exhibited cooperative evacuation behaviour in all

trials using *M. decorsei*, and only after the presentation of the blindsnake”. See revised p. 11 lines 168-169.

> Lines 195-196: Higher risk of predation – but of other predators, not the blindsnake, right? Please rephrase to be more clear.

This has been changed to “they face a higher risk of attack by predators other than blindsnakes”. See revised p. 14 lines 213-214.

> Lines 240-242: Please rephrase to make more clear that this idea of additional ant-snake symbioses in Madagascar is a hypothesis you are proposing here, but no experimental (and I think not even anecdotal?) evidence exists so far to support it.

We agree with your comment. We have added the sentence “although no experimental evidence exists to support their symbiosis”. See revised p. 18 lines 263-264.

> Line 258: "de l'Environnement et des Forêts"

We have changed “de L'environnement et des Forets” to “de l'Environnement et des Forêts”. See revised p. 19 line 278.

> Line 266: We

We have changed “I” to “We”. See revised p. 19 line 286.

Comments from Reviewer #2:

General:

Overall, I think this is an interesting study which makes a valuable and novel contribution to the literature. The authors show that ants (*Aphaenogaster swammerdami*) respond differently to 3 snake species: an ant-eating snake (*Madatyphlops decorsei*), a frog-eating snake (*Thamnosophis lateralis*), and a snake-eating snake (*Madagascarophis colubrinus*). Specifically, the ants respond to the ant-eating snake with alertness and evacuation of brood, to the frog-eating snake with some agonistic behaviour, but do not show any significant agonism towards the snake-eating snake. This suggests that not only have the ants evolved direct anti-predator responses (responding selectively to the ant-eating snake), but they have also evolved to not repel snakes that could

indirectly protect them from predators (by accepting the snake-eating snake). The experiments are well-thought out and the discussion is appropriate to the results presented, and raises some interesting questions about the evolution of ant-snake symbioses.

My concerns with this paper are about clarity and methodology, and can be addressed in a revision.

Sincerest thanks for your comments on our manuscript.

> In the abstract it is not clear what the roles of the three snakes are. It is clear that the blindsnake is a predator, and the *Thamnosophis* snake is a control, but it is not clear what the role of the lamprophiid snake is, it is simply referred to as occurring in active nests. The key point, its predation on the blindsnake, is not mentioned until the end of the abstract.

Thank you for your good advice. Although unknown species of blind snakes have been obtained from a stomach of *M. colubrinus* at our study site (A. Mori 2012, unpublished data), predation of our subject blind snake species, *M. decorsei*, by the snake-eating snake is yet to be confirmed. Therefore, the purpose of the present study was to investigate the discriminatory capacity of ants and their responses to snake species exhibiting different ecological interactions including predation or symbiosis. We proposed a potential mutualism between the snake-eating snake and the ants. Our future study would investigate the predation of *M. decorsei* by *M. colubrinus* and test whether the relationship between ants and snakes in Madagascar represents mutualism or commensalism. To clarify it, we have corrected abstract and introduction.

> In the introduction, it becomes clear that the lamprophiid snake's role in the study is as a contrasting predator of the first snake - but in this section the control snake is not mentioned at all, making it hard to interpret the results and work out which comparisons are the important ones. This just needs to be clearer.

We agree with you and have added the sentence “and *Thamnosophis lateralis* (a sympatric frog-eating snake as control)” and removed relevant sentences from materials & methods. See revised p. 5 lines 77-79 and p. 6 line 95-p. 7 line 99.

> Line 100. How was activity quantified? Or do you just mean the entrance to the nest was

recorded, i.e. video recorded? Please make this clearer

Because it is difficult to observe behaviour of all the ants, we only recorded and observed the behaviour of ants within the field of view of this video camera. To clarify we have added the sentence “We only observed the behaviour of ants within the field of view as it would be a challenge to analyse the behavior of all ants”. See revised p. 7 lines 106-108.

> Lines 117-124. Was extraction from the videos performed blind to the hypothesis? behaviours such as alertness and biting can be subjective to determine, risking unconscious observer bias if the hypothesis is known. As the analysis was done from video data, there are no logistic restrictions to prevent the data extraction being carried out blind to the identity of the snake, i.e. by a person unfamiliar with the hypothesis of the experiment and the various snake species. If the analysis has been performed blind, please add a note to this effect. If it has not been performed blind, the authors should address this by providing more information about the behavioural criteria used, and also discuss their effect sizes in the context of possible bias, or they should reanalyse a subset of their videos blind to check for observer bias effects.

We agree with you and have added more precise definition of ants' behaviour and comments on the effect sizes in the context of possible bias. See revised p. 8 line 127-p. 9 line 134 and p. 11 lines 164-167.

> l'm 124. Was the baseline activity level taken into consideration in these models? Please add a table of the model output.

We did not include the baseline activity to prevent the model becoming too complex. We have added the table to show output of the model. See added table 1.

> Line 129. Number of ants doing what? Is this all the ants on screen?

We have changed the sentence “the number of all the ants on screen”. See revised p. 9 lines 141-142.

> The paper would also benefit from thorough proof-reading for grammar. For example, in the first few lines of the abstract, line 17, "for" should read "of"; line 19 "worker's" should read "workers' ", line 22 "in" should read "on"; "termites" should read "termites' ", etc. I have not commented on other typographical and grammatical errors throughout the text.

Thank you for your comments. We have made the corrections as suggested. The English in this document has been checked by at least two professional editors, both native

speakers of English.